# Prediction of the Lethal Outcome of Acute Recurrent Cerebral Ischemic Hemispheric Stroke

**DOI:** 10.3390/medicina55060311

**Published:** 2019-06-25

**Authors:** Olexandr Kozyolkin, Anton Kuznietsov, Liubov Novikova

**Affiliations:** Department of Nervous Disease Zaporizhzhia State Medical University, 69035 Zaporizhzhia, Ukraine; o.kozyolkin@gmail.com (O.K.); titus3.05@gmail.com (A.K.)

**Keywords:** recurrent ischemic stroke, outcome, prognosis, NLR, glycemia, septum pellucidum, National Institutes of Health Stroke Scale (NIHSS)

## Abstract

*Background and objectives.* Stroke-induced mortality is the third most common cause of death in developed countries. Intense interest has focused on the recurrent ischemic stroke, which rate makes up 30% during first 5 years after first-ever stroke. This work aims to develop criteria for the prediction of acute recurrent cerebral ischemic hemispheric stroke (RCIHS) outcome on the basis of comprehensive baseline clinical, laboratory, and neuroimaging examinations. *Materials and Methods.* One hundred thirty-six patients (71 males and 65 females, median age 74 (65; 78)) with acute RCIHS were enrolled in the study. All patients underwent a detailed clinical and neurological examination using National Institutes of Health Stroke Scale (NIHSS), computed tomography of the brain, hematological, and biochemical investigations. In order to detect the dependent and independent risk factors of the lethal outcome of the acute period of RCIHS, univariable and multivariable regression analysis were conducted. A receiver operating characteristic (ROC) analysis with the calculation of sensitivity and specificity was performed to determine the prediction variables. *Results.* Twenty-five patients died. The independent predictors of the lethal outcome of acute RCIHS were: Baseline NIHSS score (OR 95% CІ 1.33 (1.08–1.64), *p* = 0.0003), septum pellucidum displacement (OR 95% CI 1.53 (1.17–2.00), *p* = 0.0021), glucose serum level (OR 95% CI 1.28 (1.09–1.50), *p* = 0.0022), neutrophil-to-lymphocyte ratio (OR 95% CI 1.11 (1.00–1.21), *p* = 0.0303). The mathematical model, which included these variables was developed and it could determine the prognosis of lethal outcome of the acute RCIHS with an accuracy of 86.8% (AUC = 0.88 ± 0.04 (0.88–0.93), *p* < 0.0001).

## 1. Introduction

Although stroke incidence, prevalence, mortality, and disability adjusted-life-year rates have declined over the last 20 years, the overall burden of stroke is still taking a lead position [1]. The incidence of stroke recurrence is high despite developments in primary and secondary preventive treatment. The first five-year cumulative incidence of stroke recurrence varies between 16% and 30% in Western countries [2]. Early recurrent stroke is relatively rare, but far more severe than first-ever stroke, is associated with longer duration of hospitalization and increased neurological disability, which makes this problem extremely relevant in medical, social, and economic aspects [3,4]. The importance of an early and reliable prognosis for recovery in patients with acute stroke is undisputed and crucial for both the treatment neurologist to optimize treatment and rehabilitation options as well as for the patient’s family [5,6].

Various prognostic indices derived from clinical features or patient characteristics and ancillary tests have been used to predict survival, discharge disposition, length of hospital stay, functional and vital outcomes. Multiple studies have proven that one of the factors, most widely accepted as prognostically helpful with I level of evidence, is baseline neurological status measured by the NIHSS [7,8,9,10,11]. Numerous studies identified a strong association between increasing age and unfavorable stroke outcome, which is independent of stroke severity, characteristics, or complication [12,13]. It was underlined that hyperglycemia on admission is associated with a worsened clinical outcome. Thus, according to Christensen H. and Boysen G. (2002), the increase in blood glucose was associated with stroke severity and short-term mortality, but not the outcome at three months [14]. Certain information has also accumulated about influence of computed tomography (CT) findings, time to treatment and recanalization, current smoking, atrial fibrillation (AF), statin intake before stroke, and proinflammatory activation on the course and outcome of ischemic stroke (IS) [15,16,17,18].

There is lack of researchers dedicated to the short-term prognosis of recurrent IS outcome described in the literature, so this justifies the necessity of our investigation, which aims to develop criteria for predicting the lethal outcome of the acute period of recurrent cerebral ischemic hemispheric stroke (RCIHS) based on a comprehensive clinical and paraclinical investigations.

## 2. Materials and Methods

This prospective, cohort, and comparative hospital-based study was approved by the ethics committee of the Zaporizhzhia state medical university (code №7 form 27 October 2016) and was performed in accordance with the Declaration of Helsinki.

Inform consent was obtained during admission of the patient to the hospital. If patient was able to give informed consent, it was written by patient; if patient was not able to write, then oral informed consent in the presence of an independent witness was given by patient. If the patient was unable to give informed consent because of physical limitation (aphasia or consciousness disorder) it was obtained from an available legal guardian (written form) or, if there was no available legal guardian, then the patient was enrolled in the study if consent was signed by next of kin. Patient/person who signed the inform consent had the opportunity to revoke the informed consent at any time during investigation.

### 2.1. Subjects

One hundred thirty-six patients (71 males and 65 females, median age 74 (65; 78)) with acute RCIHS were enrolled in the study. The patients’ inclusion to research was performed according to the following criteria: Men and women aged from 45 to 85 years, verified IS in the past, clinical and neuroimaging confirmations of acute RCIHS (a focal neurological deficit of sudden onset that persisted beyond 24 h as well as CT imaging indicating the presence of infarction in the area perfused by the middle cerebral artery (MCA)), atherothrombotic or cardioembolic subtypes of IS according to the Trial of Org 10172 in Acute Stroke Treatment (TOAST classification), first 24 h from RCIHS onset and signed informed consent of the patient‘s participation in the study.

The exclusion criteria were: Multiple ischemic strokes, hemorrhagic transformation of ischemic lesion, combined cerebral stroke, baseline NIHSS score ≥ 20, admission mRS score ≥ 3 (after first-ever stroke), presence of oncological and/or decompensated somatic pathology, anamnestic data of alcohol abuse, craniocerebral trauma, psychopathological syndrome.

We considered definition “recurrent cerebral ischemic hemispheric stroke” as any recurrent ischemic stroke occurring 24 h after the stroke onset in a different vascular region and any recurrent stroke occurring in the same region 21 days after the first-ever stroke. Affected hemispheres were defined as left and right hemisphere. Ipsilateral stroke was considered as RCIHS in the same vascular region, contralateral RCIHS–in the opposite vascular region regarding first stroke focus.

Lethal outcome (LO) of RCIHS was defined as stroke-induced mortality during the acute period of disease (during first 21 days of stroke).

All patients who survived after 21 days from stroke onset had a non-lethal outcome (NLO).

Neurological examination, neuroimaging, and laboratory investigations were performed at admission. The assessment of stroke severity was administered by an appropriately trained neurologist using the NIHSS. Brain CT was performed to evaluate the RCIHS localization, the infarct volume (IV), the presence of mass effect on the midline structures, the size and localization of the post-stroke cyst (a sign of a previously, first-ever IS). Infarct volume was calculated during CT examination by using the following formula: IV = (a × b × c × π)/6, where a, b, and c are the larger perpendicular diameters of the zone of hypodensity (cm). All CT evaluations were performed by the same neuroradiologist, who was blinded to the clinical results.

Upon admission, fasting blood samples were collected from the cubital vein to assess the levels of absolute white blood cells (WBC) and their subpopulations (absolute neutrophils count (ANC), absolute lymphocyte count (ALC), absolute monocyte count, (AMC)), glucose levels, prothrombin index, fibrinogen values in plasma, and the neutrophil-to-lymphocytic ratio (NLR).

### 2.2. Statistical Analyses

Statistical processing of the obtained results was performed using the software Statistica 13.0 (Software Inc., serial number JPZ804I382130ARCN10-J). Values were expressed as median and 25th–75th percentiles because of abnormally distributed variables (according to the Shapiro–Wilk criterion). The presence of intergroup differences in quantitative parameters was determined using the Mann–Whitney criterion. To assess the correlations between qualitative indicators Pearson χ2 criterion was used. In order to detect the dependent and independent risk factors of the LO of acute RCIHS, univariable and multivariable regression analysis were conducted. The predictive value of the indicators was evaluated using ROC analysis with the calculation of sensitivity and specificity. Area under curve (AUC) was calculated with 95% confidence interval (CI) for these variables and compared with each other. Probability (*p*) values of less than 0.05 were considered statistically significant.

## 3. Results

### 3.1. Baseline Data of Participants in Comparison with the Stroke Outcome

Baseline characteristics of participants are summarized in Table 1. The study included 136 patients with RCIHS, among them 111 patients were survived (62 males and 49 females, median age 72.0 (64.0; 77.0) and 25 patients (9 males and 16 females, median age 76.0 (74.0; 78.0) died within first 21 days of disease. Patients from both groups did not differ in gender structure (62 (55.86%) males and 49 (44.14%) females with NLO versus 9 (36.00%) and 16 (64.00%) respectively with LO (χ² = 3.22, *p* = 0.2679). No differences were observed in affected hemisphere (65 (58.56%) patients with NLO had RCIHS in the left hemisphere versus 10 (40.00%) patients with LO, χ² = 2.84, *p* = 0.9188). Patients with LO had tendency to more frequent right hemisphere stroke localization (15 (60.00%) versus 46 (41.44%), χ² = 2.84, *p* = 0.0919). Cardioembolic subtype of RCIHS, diabetes mellitus, and AF were diagnosed among 15 (60.00%), 8 (32.00%), and 15 (60.00%) patients with LO respectively; overall, there was no significant intergroup difference in the parameters of stroke localization, lateralization, stroke subtypes, and co-morbidity pathology. Patients who had LO were elder than patients with NLO (76 (74; 78) versus 72.0 (64.0; 77.0), *p* = 0.0073. The median of the time interval between first-ever stroke and recurrent one among patients with LO was 12 (6; 12) months, that was significantly different from patients with NLO, who had RCIHS in 21 (11; 36) months (*p* = 0.0034).

Clinical and paraclinical characteristics of the general cohort patients as well as results of comparative analysis of neurological, neuroimaging, biochemical, and hematological parameters in comparison with the RCIHS outcome are presented in Table 2.

As shown in the Table 2, patients with LO upon admission, had significant intergroup difference in the following variables: NIHSS score (14.0 (12.0; 16.0) points versus 12.0 (10.0; 13.0) points, *p* = 0.0003) infarct volume (64,4 (31.9; 78.5) mL versus 32.4 (17.9; 56.9) mL, *p* = 0.0118), septum pellucidum displacement (6.0 (3.5; 9.5) mm versus 2.5 (2.0; 4.0) mm, *p* = 0.0009), glucose serum level (7.8 (6.7; 9.6) mmol/L versus 5.9 (5.0; 7.2) mmol/L *p* = 0.0024), WBC (9.3 (7.3; 12.0) G/L versus 7.6 (6.3; 9.4) G/L, *p* = 0.0132), ANC (7.5 (6.2; 10.8) G/L versus 5.6 (4.3; 7.6) G/l, *p* = 0.0013), NLR (8.2 (5.0; 12.9) versus 4.5 (2.8; 7.0), *p* = 0.0041).

### 3.2. Univariable and Multivariable Logistic Regression Analysis and Prediction Model Development

All patients had complete data. The strategy of logistic regression analysis was presented as follows: At the first stage, using a simple (univariable) logistic regression analysis, among all studied indicators, the potential predictors of the lethal outcome of the acute RCHIS (OR 95% CI, *p* ˂ 0.05) were identified; at the second stage, the previously obtained predictors were step by step included in the multivariable logistic regression model and among all obtained models, one was chosen, which was distinguished by the greatest accuracy in the classification of observations.

The results of the univariable and multivariable logistic regression analysis are summarized in the Table 3.

Univariable logistic regression analysis revealed that age of patients (OR 95% CI 1.09 (1.03–1.16), *p* = 0.0059), baseline NIHSS score (OR 95% CІ 1.37 (1.17–1.63), *p* = 0.0003), infarct volume (OR 95% CI 1.01 (1.00–1.02), *p* = 0.0181), septum pellucidum displacement (OR 95% CI 1.67 (1.31–2.15), *p* = 0.0001), epiphysis displacement (OR 95% CI 1.55 (1.20–2.00), *p* = 0.0008), glucose serum level (OR 95% CI 1.21 (1.06–1.38), *p* = 0.0057), ANC (OR 95% CI 1.11 (1.00–1.23), *p* = 0.0473), ALC (OR 95% CI 0.92 (0.86–0.96), *p* = 0.0042), and NLR (OR 95% CI 1.09 (1.09–1.17), *p* = 0.0019) were the main variables, associated with the lethal RCIHS outcome.

To find out independent risk factors, which are associated with lethal RCIHS outcome, the multivariable logistic regression analysis was performed.

It was found, that the baseline NIHSS score (OR 95% CІ 1.33 (1.08–1.64), *p* = 0.0003), septum pellucidum displacement (OR 1.53 95% CI 1.17–2.00, *p* = 0.0021), glucose serum level (OR 95% CI 1.28 (1.09–1.50), *p* = 0.0022), and NLR (OR 95% CI 1.11 (1.00–1.21), *p* = 0.0303) were independent risk factors which influenced on the stroke outcome.

Independent variables have been integrated into a prediction model of the following form: β = 0.29 × P1 + 0.42 × Р2 + 0.25 × Р3 + 0.10 × P4-8.37, where P1—baseline NIHSS score, Р2—septum pellucidum displacement, P3—baseline glucose serum level, P4—NLR.

Based on ROC-analysis, it was found that β > −1.18 is the integral predictor of LO of the acute RCIHS (AUC = 0.88 ± 0.04 (0.88–0.93), *p* < 0.0001) with high sensitivity (80.0%) and specificity (86.5%) (Figure 1).

The prediction accuracy of the developed mathematical model was 86.8%.

## 4. Discussion

In this study, the predictors of LO and NLO were analyzed comprehensively and formed the prognostic model, which included baseline clinical, laboratory, and neuroimaging parameters.

The median of the time interval between first-ever stroke and recurrent one among patients with LO was 12.0 (6.0; 12.0) months, that was significantly different from patients with NLO, who had RCIHS in 21.0 (11.0; 36.0) months (*p* = 0.0034). This result is similar to data from many population-based studies dedicated to risk of recurrent stroke after a first-ever stroke and proved the hypothesis that the most “critical period” for recurrent stroke event with lethal outcome ranged from 30 days to 12 months [19,20,21,22]. 

Ischemic stroke patients with history of AF have repeatedly been shown to have higher baseline stroke severity, leading to greater disability and mortality compared to those without a history of AF We found out the tendency to higher fatality in patients with AF (60,00% vs. 39.64%). This result coincides with the opinion of Arboix A. et al. (2000) that AF was a clinical predictor of in-hospital mortality in IS patients [23]. The same assumption about higher risk of early death in acute ischemic stroke patients with AF was described by Saxena R et al. [24].

Based on our investigation, it was found that age of patient negatively affected the outcome of RCIHS. Our results are in agreement with the data from many studies, which proved that age is a significant risk factor of poor stroke outcome. Fonarow G. et al. (2010) reported that the mean age of ≥74 years was found to be important determining factor related to stroke recurrence [25], which was in accordance with data from our study. It was shown, that elderly patients have worse stroke outcomes and they have lower ability for recovering [26,27,28,29]. This result may be due to the fact that neuronal plasticity of the brain declines with ageing in contrast with increasing comorbidity and medical complications; particularly, older patients are more likely to have preexisting diseases and disabilities which may hinder their recovery.

We confirmed the hypothesis about the influence of ischemic stroke volume on the stroke outcome, which was in accordance with the results of many studies [30,31,32,33,34]. Some scientists have proven that the prediction of clinical outcomes after ischemic stroke can be improved by using imaging parameters, such as the location and volume of the ischemic lesion. Gerhard Vogt et al. (2012) in a statistical regression model included the initial lesion size, which was a strong and independent predictor of stroke outcome.

We also considered the influence of the volume of post-stroke cyst on the RCIHS outcome; however, it was not proven in our study.

It was marked that severe middle cerebral artery stroke, accompanied by malignant edema, leads to midline brain shift, which is the common cause of high mortality [35]. Based on both univariable and multivariable logistic regression analysis, we found that middle cerebral displacement, particularly displacement of septum pellucidum and epiphysis dislocation, were independent risk factors, associated with LO. This result is fully in line with W. T. Kimberly et al. (2018), who verified the association between midline shift and worse stroke outcome [36].

The baseline NIHSS score on admission is considered as the most important predictor of acute ischemic stroke outcome [37,38,39,40]. Our investigation also revealed that the severity of neurological deficits on admission assessed by NIHSS was an independent predictor of LO (OR 95% CІ 1.33 (1.08–1.64), *p* = 0.0003). Previously, Natalia S. Rost et al. (2016) reported that the NIHSS score appears to be a far more significant predictor than comorbidity score, which was thought to be an important outcome contributor [40].

High blood glucose levels in acute stroke patients is known to be related to initial neurological severity and poor outcome during the three months after stroke onset. Several studies reported that hyperglycemia after stroke is independently associated with poor functional outcome [41,42]. This study suggests that hyperglycemia has been significantly proven to be associated with LO. Our results are also similar to GLIAS (GLycemia in Acute Stroke) multicenter study where a high predictive value of an initial glucose serum levels in patients with IS was described [41]. Obtained findings are also in line with the Sung J. et al. (2017) study where serum glucose concentration during hospitalization of a patient with IS are considered highly informative indicator for the determination of stroke outcome [42].

We established that ANC in patients with LO was significantly higher than in patients with NLO and was a significant predictor of the stroke outcome. This result is consistent with Fang Y.-N. et al. (2017) who found out that higher neutrophil count was significantly predictive for in-hospital mortality in patients after non-atrial fibrillation-caused ischemic stroke (OR 95% CI 0.081 (0.057–0.104), *p* ≤ 0.0001) [43].

According to our data, the median NLR among patients with LO was almost two times higher than in patients with NLO (8.2 (5.0; 12.9) vs. 4.5 (2.8; 7.0), *p* = 0.0041). Based on univariable (NLR (OR 1.09 95% CI 1.09–1.17, *p* = 0.0019)) and multivariable (NLR (OR 95% CI 1.11 (1.00–1.21), *p* = 0.0303) analysis, the NLR appears to play a significant role in mortality after a recurrent stroke and was marked as independent predictor of LO. Our finding supports the results of Celikbilek A. et al., where proinflammatory activation and its influence on the stroke outcome have been described, particularly that NLR was significantly correlated with unfavorable clinical outcome in patients after IS. It has been highlighted that NLR is a simple, inexpensive, and readily available marker for prognosis in acute IS [44]. Tokgoz S. et al. (2014) demonstrated that NLR at the time of admission could be a predictor of short-term mortality regardless of infarct volume in IS patients [45]. Chamorro A. and Hallenbeck J. (2006) described participation of the inflammatory response in all pathophysiological processes of acute IS and clarified that ischemic brain could induce the release of proinflammatory cytokines and recruitment of immune cells, which represent an important mechanism of secondary progression of brain lesion [46].

In numerous studies was described that dementia negatively affects both survival and functional outcomes after stroke [47,48]. However, in present investigation we did not study the cognitive status of the patient, but we are hypothesized the negative influence of cognitive impairments on outcome of RCHIS. In the previous works it was found out that recurrent stroke is accompanied by more frequent and severe cognitive impairments which associated with higher disability [49,50]. We completely support the idea of A. Subic. et al. [51] and Henon H. et al. [52] that suggested that dementia could be associated with more severe global vascular disease, and a higher risk of complications, or could be a worsening factor when an intercurrent disease occurs, either because of a limited ability to respond to illness or because less invasive treatments are prescribed to demented patients. According to Béjot Y. et al. [53], dementia after stroke was not independently associated with an increased risk of death at one year. It could be explained by a shorter time window to evaluate survival compared with that used in previous studies that addressed this topic. We suggest that detailed neuropsychological assessment of stroke patients could be the crucial in order to estimate the prognosis and program for secondary prevention as well as to find the optimal treatments and for stroke patients when they are not considered as candidates for crucial intervention or thrombolysis.

We would like to emphasize that psychosocial characteristics of stroke patients may also be important predictors of mortality, especially among elder patients, who have RCHIS more often than young patients. It was estimated a higher risk of mortality in people suffering from post-stroke depression (PSD), considering depressive symptoms occurring in the early stage after the acute event. Subjects with early PSD had a risk of death about 1.5 higher as compared with non-depressed individuals, considering both short- and long-term mortality [54,55,56]. Taking into account the strong association between depressive disorders and stroke mortality, we assume that PSD requires periodic clinical attention in the long term that should focus on patients at highest risk so that, earlier identification of post-stroke depression could lead to better treatment, recovery, and reduction of the expense of patient care.

During making a prognosis of stroke outcome it is important to consider the comorbidity pathology, particularly, heart dysfunction, cardiac remodeling, and atrial fibrillation which may all affect the stroke outcome [57,58].

It was reported that hyperthermia appears to correlate with poor outcome in stroke patients. Fever leads to cerebral edema, potentially reducing cerebral perfusion pressure, and enlargers the volume of ischemic injury. Weimar at al. presented data that fever of more than 38 °C within three days after stroke was one of the most important predictors for functional dependence or death [59,60].

Taking into account all mentioned above, in order to make a reliable prognosis of the RCHIS outcome, it is important to include in predictive models the wide spectrum of variables such as age, time-interval between first-ever stroke and the recurrent one, level of neurological deficiency, neuroimaging and laboratory data, comorbidity pathology, cognitive impairments and depressive disorders. Based on our findings and statistical analysis, we have developed the prediction mathematical model, which contains variable parameters and may be implemented in the routine clinical work of Stroke Unit to obtained high sensitivity (80.0%) and specificity (86.5%) prognosis of RCIHS outcome upon admission.

## 5. Conclusions

Initial level of neurological deficiency, baseline glucose serum level, NLR, and septum pellucidum displacement are independent variables and have the most significant influence on the lethal outcome of the acute recurrent cerebral ischemic hemispheric stroke. The developed model can be applied in clinical practice to make an individual approach for the patient’s treatment as well as enlarge the variety of prescribed drugs or crucial intervention with involvement of other specialists in order to improve the care of patients.

## Figures and Tables

**Figure 1 medicina-55-00311-f001:**
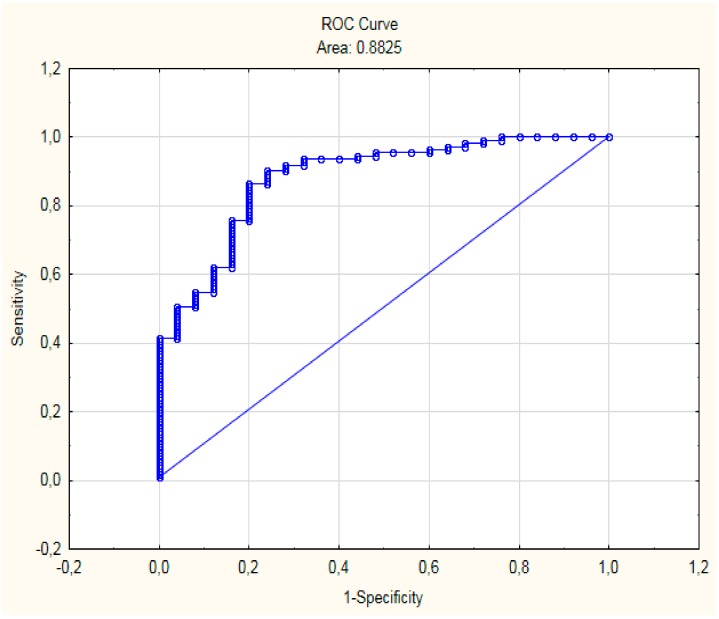
ROC-curve of predictive model.

**Table 1 medicina-55-00311-t001:** Baseline characteristics of participants in comparison with the stroke outcome.

Variables	Total (N = 136)	NLO ((N = 111)	LO	Χ^2^	*P*-Value
			(N = 25)		
Gender				3.22	0.0726
Men	71 (52.21%)	62 (55.86%)	9 (36.00%)		
Women	65 (47.79%)	49 (44.14%)	16 (64.00%)		
Lateralization of RCIHS				0.75	0.3878
Ipsilateral carotid area	61 (44.85%)	55 (49.55%)	10 (40.00%)		
Contralateral carotid area	75 (55.15%)	56 (50.45%)	15 (60.00%)		
Stroke classification according TOAST				3.44	0.0635
Atherothrombotic RCIHS	77 (56.62%)	67 (60.36%)	10(40.00%)		
Cardioembolic RCIHS	59 (43.38%)	44 (39.64%)	15 (60.00%)		
Affected hemisphere				2.84	0.0919
Left	75 (55.15%)	65 (58.56%)	10 (40.00%)		
Right	61 (44.85%)	46 (41.44%)	15 (60.00%)		
Diabetes mellitus	31 (22.79%)	23 (20.72%)	8 (32.00%)	1.48	0.2246
Atrial fibrillation	59 (43.38%)	44 (39.64%)	15 (60.00%)	3.44	0.0635

**Table 2 medicina-55-00311-t002:** Clinical and paraclinical characteristics of general cohort of patients with recurrent cerebral hemispheric ischemic stroke and comparative analysis of clinical neurological, computed tomography, biochemical, and hematological parameters in comparison with the outcome of acute period of disease.

Variables, (Me (Q1; Q3)	Total ((N = 135)	NLO ((N = 111)	LO ((N = 25)	*P*-Value
NIHSS score at baseline, points	12.0 (10.0; 14.0)	12.0 (10.0; 13.0)	14.0 (12.0; 16.0)	0.0003
Infarct volume, mL	35.7 (22.4; 65.3)	32.4 (17.9; 56.9)	64,4 (31.9; 78.5)	0.0118
The volume of the post-stroke cyst, mL	9.2 (1.9; 19.8)	8.3 (2.1; 18.4)	12.9 (3.7; 56.3)	0.4224
Septum pellucidum displacement, mm	3.5 (2.0; 6.0)	2.5 (2.0; 4.0)	6.0 (3.5; 9.5)	0.0009
Epiphysis displacement, mm	3.0 (2.0; 4.0)	3.0 (3.0; 4.0)	3.5 (2.0; 4.5)	0.9798
Glucose serum level, mmol/L	6.13 (5.0; 7.84)	5.9 (5.0; 7.2)	7.8 (6.7; 9.6)	0.0024
Fibrinogen, g/L	3.5 (2.9; 4.4)	3.5 (2.9; 4.4)	3.5 (3.3; 4.2)	0.4699
Prothrombin index, %	90.0 (85.5; 95.0)	90.0 (86.0; 96.0)	88.0 (86.0; 94.0)	0.5392
Hematocrit, %	41.0 (37.5; 45.0)	41.0 (37.5; 45.0)	41.0 (38.0; 45.0)	0.8460
White blood cells, G/L	7.8 (6.4; 10.2)	7.6 (6.3; 9.4)	9.3 (7.3; 12.0)	0.0132
Absolute neutrophil count, G/L	6.1 (4.5; 8.2)	5.6 (4.3; 7.6)	7.5 (6.2; 10.8)	0.0013
Absolute lymphocyte count, G/L	1.3 (0.8; 1.9)	1.3 (0.9; 1.9)	0.9 (0.7; 1.6)	0.0861
Absolute monocyte count, G/L	0.4 (0.3; 0.6)	0.4 (0.2; 0.6)	0,4 (0.3; 0.6)	0.5534
Neutrophil-to-Lymphocyte Ratio	4.8 (2.9; 8.1)	4.5 (2.8; 7.0)	8.2 (5.0; 12.9)	0.0041

**Table 3 medicina-55-00311-t003:** The results of logistic regression analysis.

Variables	Univariable Logistic Regression Model		Multivariable Logistic Regression Model	
	OR (95% CI)	*P*	OR (95% CI)	*P*
Age, years	1.09 (1.03–1.16)	0.0059		
NIHSS score at baseline, points	1.37 (1.17–1.63)	0.0003	1.33 (1.08–1.64)	<0.0001
Infarct volume, mL	1.01 (0.01–1.02)	0.0181		
Septum pellucidum displacement, mm	1.67 (1.31–2.15)	0.0001	1.53 (1.17–2.00)	0.0021
Epiphysis displacement, mm	1.55 (1.20–2.00)	0.0008		
Glucose serum level, mmol/L	1.21 (1.06–1.38)	0.0057	1.28 (1.09–1.50)	0.0022
Absolute neutrophils count, G/L	1.11 (1.00–1.23)	0.0473		
Absolute lymphocyte count, G/L	0.92 (0.86–0.97)	0.0042		
Neutrophil-to-Lymphocyte Ratio	1.09 (1.02–1.17)	0.0119	1.11 (1.00–1.21)	0.0303

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
