# Peer review of "Prediction of the Lethal Outcome of Acute Recurrent Cerebral Ischemic Hemispheric Stroke"

_1010-660X, 2019, doi:10.3390/medicina55060311_

Round 1

Reviewer 1 Report

Manuscript ID: medicina-503576
Type of manuscript: Article
Title: Prediction of the lethal outcome of acute recurrent cerebral ischemic hemispheric stroke

Authors: Olexandr Kozyolkin, Anton Kiznietzov, Liubov Novikova *

This is an interesting study about to investigate the criteria for predicting the lethal outcome of RCIHS. In spite of theses attractive results, some careful considerations should be made.

1.      I wonder what your aim of this study is.

The variables you suggested (NIHSS, infarct volume, midline shift, serum glocuse, and NLR… ) were already well-known. So there is no new added information from this article.

2.      Do you want to make a scoring system for predicting RCIHS?

If so, the sample size was too small to make a predicting model.

3.      Based of the previous reports, cardioembolic etiolgy and AF were well known risk factors for poor outcomes. Maybe it might be related to small sample size.

4.      What is “lethal outcome” or “Non-lethal outcome”?

Mortality? Severe initial neurological severity? 3month mRS ?

There are no informations about those definitions. Please clarify.

Author Response

We appriciate your independent review of our work and would like to give the following answers:

1. We agree with your comment, in the literature exists information about prognosis of first-ever stroke outcome, however  the data on outcome of recurrent strokes has poorly studied, so that we aimed to develop criteria for predicting the lethal outcome of the recurrent stroke based on comprehensive clinical and paraclinical data, obtained upon admission. The developed multivariable model allows predicting the lethal outcome of the acute RCHIS based on a comprehensive analysis of four the most informative predictors (baseline NIHSS score, septum pellucidum displacement, baseline serum glucose level, NLR). These four predictors were integrated into the binary logistic regression equation of the following type: β = 0.42 * (septum pellucidum displacement) + 0.25 * (baseline serum glucose level) + 0.10 * NLR + 0.29 * (baseline NIHSS score) -8.37. Based on ROC analysis, the optimal beta value (-1.18) was determined, which acts as an integral predictor of the lethal outcome of the acute RCHIS (sensitivity = 80.0%, specificity = 86.5%). 

2. We are interested in making a scoring system for predicting RCIHS in our future researches, and still collecting data for this. The idea of present work was to develop the prediction criteria of the lethal outcome of recurrent stroke. The following criteria were used to assess the informative content of the developed multivariable logistic regression model: 1) the accuracy of the classification of observations (86.8%); 2) AUC (0.88, p˂0.05); 3) model fits test p˂0.05). The foregoing substantiates the appropriateness of using this model in routine clinical practice to predict the lethal outcome of the acute RCHIS because it fits test p˂0.05

3. We found out the tendency to higher fatality in patients with AF (60,00% vs. 39.64%) However, in this study, the presence of AF did not demonstrate an independent association with the risk of lethal outcome of acute RCHIS, which, in our opinion, was due to the smaller effect of AF on the risk of death in comparison with indicators such as baseline NIHSS score, septum pellucidum displacement, serum glucose level, NLR.

4.  Explanations added to the article: 

Lethal outcome (LO) of RCIHS was defined as stroke-induced mortality during the acute period of disease (during first 21 days of stroke). All patients who survived after 21 days from stroke onset had a non-lethal outcome (NLO).

Thank you! 

Reviewer 2 Report

I have the following comments that could hopefully improve the manuscript:

1) Abstract: contains too many abbreviations, which makes it hard t read.

2) Ethical aspects: please indicate whether the patients gave an informed consent to participate in this study and provide with more details, such as at which state the patients were when they gave the consent, whether they had a right to withdraw their data etc

3) Analysis: Multivariate – I assume you mean multivariable? Please indicate whether all your patients had complete data. If now, please state missingness and how you handled them

Covariates entered into the multivariable regression model: please describe your analytical strategy. First, provide a rationale why you choose to study all these clinical variables as predictors into the model. Given the large number of compared variables (Table 2), some of the p values may be just by chance. Second, please state explicitely which variables you entered into the final model. Also please discuss entering all these predictors given your sample size. Present results of the multivariable analysis in a table.    

4) Discussion: please take into account cognitive status of the patients. If there is no data on it, at least discuss it in line with literature, for example follwing papers could be of use: Management of acute ischaemic stroke in patients with dementia by Ana Subic in Journal of Internal Medicine, Treatment of atrial fibrillation in patients with dementia: A cohort study from the Swedish Dementia Registry by Ana Subic in Journal of Alzheimer´s Disease

Please discuss other factors that may have played a role, but you did not have data on them. For example depressive symptoms are strong predictors of mortality and they are very common in older adults from Central and Eastern Europe (see for example Horackova et al., Prevalence of late-life depression and gap in mental health service use across European regions in European Psychiatry), even though Ukraine did not participate in this European study, I assume depressive symptoms of older adults are high and would be relevant for your study.

In the discussion, please omit repetition of results, instead, please sum up your results and discuss in light with literature. Please add what your results mean in a wider context, such as how your results can be used to improve care of patients.  

5) There are several typos and English proofreading is needed – for example „there is lack of researches“ etc, „were survived“, „from both group“, „reviled“. Abbreviations: please explain them the first time you are using them, such as TOAST

Author Response

We appriciate your independent review of our work and would like to give the following answers:

1. The abstact was corrected, please see attached file. 

2. Inform consent was obtained during admission of the patient to the hospital. If patient was able to give inform consent, it was written by patient, if patient was not able to write, then oral inform consent in the presence of independent witness was given by patient. If patient was unable to give the inform consent, because of physical limitation (aphasia or consciousness disorder) it was obtained from available legal quardian (written form) or, if no available legal quardian, then the patient was enrolled in the study if consent was signed by next of kin. Patient/Person who signed the inform consent had the opportunity to reward the inform consent at any time during investigation.

3.Multivariable analysis was performed. All patients had complete data. The strategy of logistic regression analysis was presented as follows: at the first stage, using a simple (univariable) logistic regression analysis, among all studied indicators, potential predictors of the lethal outcome of the acute RCHIS were identified (OR 95% CI p˂0.05); at the second stage, the previously obtained predictors were step by step included in the multivariable logistic regression model and among all obtained models, it was chosen one, which was distinguished by the greatest accuracy in the classification of observations (86.8%). The following independent predictors were entered in the final multivariable model: baseline NIHSS score, septum pellucidum displacement, serum glucose level, NLR.

4. Thank you! The new data entered in the discussion

In numerous studies was described that dementia negatively affects both survival and functional outcomes after stroke [47, 48]. However, in present investigation we did not study the cognitive status of the patient, but we are hypotized the negative influence of cognitive impairments on outcome of RCHIS. In the previous works it was found out that recurrent stroke accompanies by more frequent and severe cognitive impairments which associated with higher disability [49, 50]. We completely support the idea of A. Subic. et al [51] and Henon H. et al. [52] who suggested that dementia could be associated with more severe global vascular disease, and a higher risk of complications, or could be a worsening factor when an intercurrent disease occurs, either because of a limited ability to respond to illness or because less invasive treatments are prescribed to demented patients. According to Béjot Y. et al. [53] dementia after stroke was not independently associated with an increased risk of death at 1 year. It could be explained by a shorter time window to evaluate survival compared with that used in previous studies that addressed this topic. We suggest that detailed neuropsychological assessment of stroke patients could be the crucial in order to estimate the prognosis and program for secondary prevention as well as to find the optimal treatments and for stroke patients when they are not considered as candidates for crucial intervention or thrombolysis.

We would like to emphasize that psychosocial characteristics of stroke patients may also be important predictors of mortality, especially among elder patients, who have RCHIS more often than young patients. It was estimated a higher risk of mortality in people suffering from post-stroke depression (PSD), considering depressive symptoms occurring in the early stage after the acute event. Subjects with early PSD had a risk of death about 1.5 higher as compared with non-depressed individuals, considering both short- and long-term mortality [54-56]. Taking into account the strong association between depressive disorders and stroke mortality, we assume that PSD requires periodic clinical attention in the long term that should focus on patients at highest risk. So that, earlier identification of post-stroke depression could lead to better treatment, recovery and reduce the expense of patient`s care.

During making a prognosis of stroke outcome it is important to consider the comorbidity pathology, particularly, heart dysfunction, cardiac remodeling and atrial fibrillation which may all affect the stroke outcome [57, 58].

It was reported that hyperthermia appears to correlate with poor outcome in stroke patients. Fever leads to cerebral edema, potentially reducing cerebral perfusion pressure, and enlargers the volume of ischemic injury. Weimar at al. presented data that fever of more than 38 °C within three days after stroke was one of the most important predictors for functional dependence or death [59, 60].

5. Thank you! Corrections have been done, please see the attached file.

Round 2

Reviewer 1 Report

Medicina (ISSN 1010-660X)

Manuscript ID : medicina-503576

Title: Prediction of the lethal outcome of acute recurrent cerebral ischemic hemispheric stroke

Authors: Alexandr Kozyolkin , Anton Kiznietzov , Liubov Novikova *

1. I still have to present the methodological issue of this study.

Some well known predictors for poor outcomes in stroke (AF, dysphasia, low BMI… ) were not significant or not used.

References>

Prediction of Recurrent Stroke or Transient Ischemic Attack After Noncardiogenic Posterior Circulation Ischemic Stroke. Stroke. 2017;48:1835–1841

Prediction of Recurrent Stroke and Vascular Death in Patients With Transient Ischemic Attack or Nondisabling Stroke. Stroke. 2010;41:487–493

A score to predict early risk of recurrence after ischemic stroke. Neurology. 2010 Jan 12; 74(2): 128–135.

Predictors of Recurrent Stroke in Patients with Ischemic Stroke: Comparison Study between Transesophageal Echocardiography and Cardiac CT. Radiology https://doi.org/10.1148/radiol.15142300

Author Response

We are gratuful for your through review of our work and would like to give the answer on yor question:

In the available literature, the effect of these factors (dysphasia, low BMI, AF) on the risk of the lethal outcome of recurrent stroke, which has already occurred, have not clearly described. 

The specific objective of our study was to determine the predictors of a short-term mortality (death within 21 days from the development of the RCIHS) among patients with occurred recurrent stroke (all patients have first-ever stroke before and were admitted to the hospital with recurrent ischemic stroke). The observation period for patients was 21 days. The independent presictors of the  lethal outcome (short-term mortality) of recurrent stroke were: baseline NIHSS score, septum pellucidum displacement, glucose serum level and NLR. 

Medicina EISSN 1010-660X Published by MDPI AG, Basel, Switzerland RSS E-Mail Table of Contents Alert
Back to Top